# *Leptospira* Detection in Cats in Spain by Serology and Molecular Techniques

**DOI:** 10.3390/ijerph17051600

**Published:** 2020-03-02

**Authors:** Andrea Murillo, Rafaela Cuenca, Emmanuel Serrano, Goris Marga, Ahmed Ahmed, Salvador Cervantes, Cristina Caparrós, Verónica Vieitez, Andrea Ladina, Josep Pastor

**Affiliations:** 1Department de Medicina i Cirurgia Animals, Facultat de Veterinària, Universitat Autònoma de Barcelona, Barcelona (UAB), CP 08193 Bellaterra, Espana; rafaela.cuenca@uab.es (R.C.); Josep.Pastor@uab.cat (J.P.); 2Wildlife Ecology & Health group (WE&H), Servei d’Ecopatologia de Fauna Salvatge (SEFaS), Universitat Autònoma de Barcelona (UAB), CP 08193 Bellaterra, Espana; emmanuel.serrano@uab.cat; 3OIE and National Collaborating Centre for Reference and Research on Leptospirosis (NRL), Amsterdam UMC, University of Amsterdam, Medical Microbiology, Meibergdreef 39, 1105 AZ Amsterdam, The Netherlands; m.goris@amsterdamumc.nl (G.M.); a.ahmed@amsterdamumc.nl (A.A.); 4Clínica Felina Barcelona, CP 08015 Barcelona, Espana; clinicafelinabarcelona@gmail.com (S.C.); cristinacapa94@gmail.com (C.C.); 5Facultad de Veterinaria, Universidad de Extremadura, CP 10003 Cáceres, Espana; vvieitez@gmail.com (V.V.); rabacha@unex.es (A.L.)

**Keywords:** antibodies, free-roaming cat, leptospirosis, PCR, shedding, serovar, zoonoses

## Abstract

Leptospirosis is the most neglected widespread zoonosis worldwide. In Spain, leptospirosis reports in people and animals have increased lately. Cats can become infected with *Leptospira*, as well as be chronic carriers. The aim of this study was to determine serological antibody prevalence against *Leptospira* sp., blood DNA, and shedding of DNA from pathogenic *Leptospira* species in the urine of cats in Spain. Microagglutination tests (MAT) and blood and urine TaqMan real-time polymerase chain reaction (PCR) were performed. *Leptospira* antibodies were detected in 10/244 cats; with 4.1% positive results (95% confidence interval (CI): 2.1–7.18%). Titers ranged from 1:20 to 1:320 (serovars Ballum; Bataviae; Bratislava; Cynopteri; Grippotyphosa Mandemakers; Grippotyphosa Moskva; Pomona; and Proechimys). The most common serovar was Cynopteri. Blood samples from 1/89 cats amplified for *Leptospira* DNA (1.12%; 95% CI: 0.05–5.41%). Urine samples from 4/232 cats amplified for *Leptospira* DNA (1.72%; 95% CI: 0.55–4.10%). In conclusion free-roaming cats in Spain can shed pathogenic *Leptospira* DNA in their urine and may be a source of human infection. Serovars not previously described in cats in Spain were detected; suggesting the presence of at least 4 different species of pathogenic leptospires in the country (*L. borgpetersenii*; *L. interrogans*; *L. kirschneri*; *and L. noguchii*).

## 1. Introduction

Leptospirosis is a zoonosis present in every continent except Antarctica. The vast majority of mammals have been shown to be hosts of *Leptospira* [1,2]. Worldwide, 1.03 million cases of *Leptospira* infections in humans have been estimated per year, of which 58,900 correspond to deaths [3]. The incidence of human leptospirosis in Spain is 0.86 cases per million inhabitants. Catalonia and Extremadura are two of the autonomous communities with the highest reported cases [4,5]. It has been shown that some pathogenic species of leptospires like *Leptospira borgpetersenii*, *Leptospira interrogans*, *and Leptospira kirschneri* can naturally infect cats [6,7,8]. Clinical presentation of the disease is rare and usually mild in cats [9,10,11,12]. Several studies in different geographical areas have demonstrated that cats have contact with *Leptospira* since they develop specific antibodies ranging from 4% to 33.3%, with no clear association to clinical disease [13,14,15,16,17]. Urinary shedding of *Leptospira* DNA has also been documented in cats, with a prevalence ranging up to 67.8% depending on various factors including the geographical area, the presence in the area of farm animals infected with leptospires, and prey habits [7,14,15,18,19,20,21]. Furthermore, a recent study investigated the ability of cats to excrete viable bacteria through urine (p. 227, [19]), suggesting the possibility that cats can spread the bacteria by urine and infect humans. Sequences from bacterial DNA isolated in acute human cases of leptospirosis and wild animals have shown similarities with those isolated from cats in Reunion Island *(Leptospira borgpetersenii)*; the authors of the work rejected, however, any major role of feral cats in the epidemiology of leptospirosis in that geographical region due to urinary *Leptospira* shedding cats extremely low prevalence (0.6%) [8]. Conversely, other researchers argue that the role of cats in the maintenance of the pathogen has so far been underestimated [6,15,19]. It is possible that cats have an important role as leptospires carriers, contributing to their maintenance in the environment and favouring the zoonotic risk. The aim of this study was to evaluate the presence of antibodies against pathogenic *Leptospira* species and to determine the presence of *Leptospira* DNA in urine and blood by means of polymerase chain reaction (PCR), in free-roaming cats from two different geographical areas in Spain.

## 2. Materials and Methods

Sample size was determined based on the formula (n = Z² × P × (1−P) /d² where *n* = required sample size; *Z* = Z statistic for a level of confidence; *P* = expected prevalence or proportion based on literature in proportion of one; and *d* = precision in proportion of one), proposed for prevalence studies. Based on an assumed prevalence of antibodies against *Leptospira* species in cats of 17.9% and leptospiral DNA shedding in urine of 3.3% [15], a sample size of 226 cats (95% CI; 5% precision) was required for antibody prevalence and 194 cats (99.9% CI; 5% precision) for DNA urinary shedding study.

### 2.1. Animals in the Study

Animals from two different geographical regions of Spain (Barcelona, Catalonia, in the northeast and Cáceres, Extremadura, in the southwest) were used in this prospective trial. Two hundred and forty-four cats were recruited from October 2017 to September 2018. Cats from Barcelona (90/244) were part of a neutering program and were housed in local animal shelters in Barcelona; cats from Extremadura (154/244) were part of a free-roaming cat spay program in Cáceres. Owing to the feral nature of most cats, prior history was not available. Data for gender, estimated age, and breed were collected. Signed informed consent was obtained from all animal shelters. Sampling collection was performed under the guidelines of the Ethical Committee Animal Care and Research, Autonomous University of Barcelona, approval number CEEAH, code 2939.

### 2.2. Sample Collection

In all cases, samples were collected with the cat under general anaesthesia. Three ml of venous jugular blood, collected from each animal, was divided and 1 ml was transferred to a K3-EDTA tube and 2 ml to a serum separator tube. K3-EDTA tubes were frozen at −20 °C until the DNA extraction. Serum separator tubes were centrifuged at 1300 × *g,* 5 minutes, within 5 hours of collection and the serum was stored at −20 °C until testing for leptospiral antibodies. Urine was collected by direct cystocentesis in sterile syringes and centrifuged at 14,000× *g,* 15 minutes at room temperature. The supernatant was discarded, and the pellet was re-suspended in a ratio of 1:1 with 0.5 ml Buffer phosphate-buffered saline (PBS) (pH 7.6, 0.01 M, Canvax, Córdoba, Spain). The final pellet volume (1ml) was stored at −20 °C until DNA extraction was performed. All cats were tested for feline leukaemia virus, FeLV and feline immunodeficiency virus, FIV, using a commercial test (SNAP Combo FeLV/FIV test®, IDEXX, Barcelona, Spain).

### 2.3. Microscopic Agglutination Test (MAT)

Microscopic agglutination test (MAT) was performed at the OIE and National Collaborating Centre for Reference and Research on Leptospirosis, Amsterdam, the Netherlands. MAT was performed by direct reading following a technique described before [22]. When a reaction was observed, it was re-checked by indirect reading. Two-fold serial dilutions of serum from 1:20 to 1:160 were tested; this is including the antigen. In the case of animals with titers ≥1:160, the test was repeated, with a serial dilution series from 1:20 to 1:10240. Any antibody titer ≥1:20 was considered as a positive value [23]. Serum was examined for antibodies against 8 species of *Leptospira*, belonging to 20 serogroups, 27 serovars, and 28 strains (Table 1). Saprophytic strains *L. biflexa* strain Patoc I, *L. biflexa* strain CH 11, and *L. meyeri* strain Veldrat Semarang 173, were included in our panel diagnostic according to the World Health Organization’s guidance [24].

### 2.4. DNA Isolation

Total nucleic acid extraction from blood and urine samples was performed using The NucliSENS EasyMAG® automated system (bioMérieux, Marcy l’Etoile, France) according to the manufacturer’s instructions. Blood samples (K3-EDTA 1 ml tube), were suspended in 2 ml EasyMAG lysis buffer. Urine samples were centrifuged at 14,000× g 30 minutes at room temperature, the supernatant was discarded and the pellet was re-suspended in 2 ml EasyMAG lysis buffer. The DNA was eluted in 80 µl elution buffer in the last step of the extraction procedure.

### 2.5. TaqMan Real-Time PCR

DNA extracted from cats’ biological materials were tested with TaqMan real-time PCR described by Ahmed et al., 2020 [25]. Briefly, primers and probe sequences targeting *lipL32* gene-specific for pathogenic *Leptospira* (LipgrF2, LipgrR2, and LipgrP1) and the internal set primers, probe, and synthetic internal control template sequences (IntoF2, IntoR2, IntoP1, and PlasintS1) were listed in Table 2. The PCR analytical sensitivity for the spiked serum, blood, and urine with *L. interrogans* were estimated as 2, 3, and 5 leptospires per reaction respectively. The PCR as described has a high specificity and is capable of detecting all pathogenic *Leptospira* so far known. Between 100–500 copies per reaction of genomic DNA extracted from *Leptospira interrogans* strain Kantorowic was used as a positive control. The PCR was performed, including the internal control template to monitor the reaction performance and double-distilled DNase/RNase-Free water as a negative control. All clinical samples were tested in duplicate.

The concentration of the reagents and the cyclic amplification protocol were the following; 12.5 ul of 2x master mix (applied biosystem), 0.4 µM of each leptospires forward and reveres primer (LipgrF2 and LipgrR2), 0.2 µM of the leptospires probe (LipgrP1), 0.16 µM of each internal control primers (IntoF2 and IntoR2), 0.08 µM of internal control probe (IntoP2), 0.25 µl double-distilled DNase/RNase-free water and 0.29 pg (equivalent to 50 copy) of internal control DNA template (PlasintS1). Finally, 10 µl of DNA extracted from cats’ DNA in a total volume of 25 µl were submitted to an amplification procedure using the CFX96 real-time PCR detection system (Bio-Rad, Amsterdam, The Netherlands). The cyclic amplification protocol consists of the following steps; initial DNA denaturalization and DNA polymerase activation at 95 °C for 5 minutes, 45 cycles of two steps, 95 °C for 20 seconds as denaturalization and 60 °C for 30 seconds as hybridization, and annealing-extension steps for the probes and each forward and reverse primers respectively.

### 2.6. Statistical Analysis

Apparent prevalence, 95% CI and CI of a proportion were calculated for antibodies against *Leptospira* serovars, leptospiral DNA in blood, leptospiral DNA urinary shedding, FeLV, and FIV infection with OpenEpi (Andrew G. Dean and Kevin M. Sullivan, Atlanta, GA, USA) [26]. Descriptive statistics were performed for the calculation of medians, mean, SD, and range. For the possible risk factors for *Leptospira* infection, the linearity assumption was first guaranteed with multiple logistic regression. Age, gender, sampling season, and co-infections with FeLV/FIV were analysed for binary logistic regression as possible risk factors associated with *Leptospira* (antibodies presence, amplified *Leptospira* DNA in blood and/or urine). A *P* value <0.05 was determined as statistically significant. Statistical analysis was performed using a commercial software program (IBM SPSS-Statistics version 22, IBM © Armonk, NY, USA).

## 3. Results

In total, 90/244 cats were from Barcelona and 154/244 were from Cáceres, all were domestic short-haired, 131 cats were male and 113 were female, with ages from 3 months to 16 years (mean 1.8 years and SD 2.30). In addition, 17/244 were FeLV positive (7%, 95% CI: 4.24–11%) and 7/244 were FIV positive (2.9%; 95% CI: 1.3–5.6%). None of the animals were positive for both diseases (FeLV/FIV). One hundred and thirty-five cats were sampled in winter, 58 in spring, 15 in summer, and 36 in autumn.

### 3.1. Seroprevalence

Serum was obtained from all animals and 10/244 cats (4.1–95% CI: 2.1–7.18%), 9 from Cáceres and only one from Barcelona; 8 males and 2 females with ages ranged from 6 months to 6 years, were seropositive (antibody titers ranged from 1:20 to 1:320) for at least one serovar. Only two cats had antibody titers ≥ 1:320 against the serovars Bataviae and Proechimys (Table 3).

The most common serovars involved in the study were Cynopteri (5/10 of seropositive cats) followed by serovars Ballum, Bratislava, Grippotyphosa, and Proechimys. Antibodies titers for at least two serovars belonging to two different serogroups were detected in 2 cats. None of the cats with antibodies shed pathogenic *Leptospira* DNA in their urine. All animals were negative against saprophytic strains included in the panel.

### 3.2. DNA Detection in Blood PCR

Due to the low volume of the blood sample obtained during sampling in some animals, DNA isolation from blood samples was only possible in 89/244 cats. Only one sample (8-month-old female from Barcelona), was positive (1.12%; 95% CI: 0.05–5.41%). This cat had no antibodies against *Leptospira* detected by MAT (Table 4).

### 3.3. Urinary Shedding

It was not possible to collect urine in 12 cats and therefore, 232 urine samples were processed for DNA extraction and subsequently PCR testing. A total of 4/232 samples amplified DNA from pathogenic *Leptospira* species (1.72%; 95% CI: 0.55–4.10%); two cats (1 male and 1 female) were from Cáceres and two cats (1 male and 1 female) were from Barcelona. All positive cats were ≥1-year-old (Table 4). All PCR negative controls tested negative and all PCR positive controls tested positive. None of these cats had antibodies against *Leptospira* by MAT.

### 3.4. Risk Factor Analysis

Multivariate logistic regression did not reveal significant risk factors for *Leptospira* infection neither for seroprevalence nor DNA detection in blood and/or urine in the present study; *P* values were ≥0.05.

## 4. Discussion

Cats are susceptible to *Leptospira* [9,10,12,27] and the presence of viable pathogenic leptospires in the urine of cats has been proven (p. 227, [19,28]). Therefore, the species can play a role in the transmission of the zoonosis. Cats are becoming more popular as a companion animal and it is therefore important to have data to assess the extent to which cats constitute a risk of human leptospiral infection.

Scant information is known about specific characteristics of leptospirosis pathogenesis in cats. Based on general knowledge of the disease, once an animal becomes infected it may develop the incidental host state with the presentation of acute illness, that may be fatal, or chronic renal carrier state with mild or non-presenting clinical signs [23]. There are some differences in the disease presentation linked to the infecting serovar [29]. Leptospires enter the warmer body environment and transcriptional changes occur that enhance their pathogenicity [23]. In rats and mice (chronic carrier host), the regulation of *Leptospira* lipopolysaccharide (LPS) differs from human and dogs (incidental host) [30,31]. In murine species, there is an adaptation of the innate response to infection with leptospires [32]. In carrier hosts (rats and mice), leptospires are disseminated through the organism and are most likely cleared by the immune system from all tissues except the kidney. In the epithelial cells of the renal tubules, leptospires continue to multiply and are shed in the urine [33].

Based on experimental infections and previous reports on cats, leptospiraemia may be present in the first hours of infection [34], but on average, it appears from 6 days post-infection and lasts up to 7 more days [35]; antibody titer rises at the end of the first week of infection [34], the peak titer has been reported to be around day 21 [35] but in many cases, cats unlike dogs, do not develop a high antibody titer [34,35,36]. In other species, antibodies last from months to years [2,23], but it has not yet been confirmed in cats. The shedding of leptospires in cats’ urine appears from 2–4 weeks post-infection and it could last at a maximum of 6 weeks in case of acute disease (incidental host state) [34,35]. Acute cases of feline leptospirosis, however are scarce nevertheless, epidemiological studies on leptospires prevalence demonstrate that the role of cats is mostly as a chronic carrier. In a cat, urinary shedding of pathogenic *Leptospira* has been demonstrated for a period of 8 months [15]. Based on the above information, it is our belief that cats as the murine species act most as the chronic carrier than an incidental host for the disease.

Seroprevalence against *Leptospira* observed in our study (4.1% positive 95% CI: 2.1–7.18%), fell within the previous intervals described worldwide, 4% to 33.3% [14,16,17,37]. Environmental factors such as outdoor habits, presence of farm animals that may shed leptospires in the neighborhood, prey habits, or even the season of the year (resulting in different levels of exposure to pathogenic leptospires), can explain the broad ranges of antibody prevalence reported in the literature. Even different cut off values (≥1:100) and serovar panels used in laboratories may affect the prevalence. All these factors, along with a different length of sampling (3 years) and sample size (*n* = 53), may explain the prevalence (14%) obtained previously in the country [38], compared with the lower prevalence in our study (4.1%).

Except for two cats from Cáceres (titer 1:320), antibody titers of the animals in our study were not ≥1:100. Adler, 2014 [23], reported that infected animals may have MAT titers below 1:100; which is supported by epidemiological studies in cats [13,14,16,39]. Seropositive animals in our study demonstrate previous evidence of contact with *Leptospira*. They could have had a recent infection, as cats are not as routinely vaccinated against the disease as dogs, or they were chronically infected animals with falling antibody titers.

At the time of sampling, both cats with titers ≥1:320 showed no urinary shedding of *Leptospira* DNA. Nevertheless, we were unable to take further urine samples from these animals at different times (as per Weis et al., 2017) [15] in order to confirm if they had intermittent shedding of the bacteria DNA. To our knowledge, this is the first study in Spain to confirm by MAT, the seropositivity against serovars Bataviae, Bratislava, Cynopteri, Grippotyphosa, Pomona, Proechimys, and Rachmati in cats, suggesting the possible presence of serovars from serogroups Australis, Autumnalis, Bataviae, Cynopteri, Grippotyphosa, and Pomona among cats in the country. In the metropolitan area of Barcelona, Spain, the presence of members of serogroups Australis, Bataviae, and Grippotyphosa (also detected in our study in cats) has been detected in small mammals [40], so it may be possible for serovars to circulate between animal species. Another factor to consider is the increased contact between cats and animal reservoirs of leptospires, due to shifts in the dynamics and colonization of cities. Environmental changes often increase the frequency and magnitude of contact between wild and domestic species, thus increasing the risk of disease transmission. Leptospirosis is an example of this dynamic interface with wildlife [41]. Serovars belonging serogroups Australis, Autumnalis, Batavie, Bratislava, Grippotyphosa, and Pomona have been previously described in cats from Europe [12,15,16,27,39,42].

A cross-sectional epidemiological *Leptospira* study was recently conducted in dogs from Spain [43]; however, serovars from serogroups Ballum, Bataviae, and Cynopteri, were not part of the diagnostic panel, unlike our study of cats which included them. Antibodies against serovars of these serogroups were present in the cats of our study. The variety and number of serovars and serogroups included in the diagnostic panel have a direct relationship with the sub-diagnosis of leptospirosis by MAT [44].

Generally speaking, cross-reactivity between leptospiral serogroups has been previously described. In dogs, antibody titers to heterologous strains may provide equal or higher titers than the infecting serovar [23]. A second test performed 15 days after the first one, could help determine the infecting serovar due to seroconversion. Although, still caution is needed since presumptive serogroup data should be used only to give a broad idea of the common serogroups present in a population and cannot be interpreted reliably in individual patients [45]. In the present study, 3 cats had antibody titers against more than one serovar (Table 3). Previously published studies [15,20,46] have also reported cats simultaneously seropositive against different serovars. The simultaneous seropositivity that some cats display, not only in our study but in others, could be explained by either a genuine cross-reactivity in cats or by the simultaneous exposure of the animals to different serovars belonging to different serogroups. In the case of the one-year-old female cat from Cáceres, she had antibodies against 7 different serovars. Surely in the case of the serovars belonging to serogroup Pomona, it represents a cross-reaction with the highest titer obtained.

Cynopteri (belonging a Cynopteri serogroup) was the most frequently detected serovar, in our study. In all cases, the infected animals came from Cáceres; therefore, there is a possibility that a serovar from this serogroup is endemic in the cat population of this region. None of the seropositive cats shed *Leptospira* DNA in their urine. A plausible explanation for this could be that either the animals had an acute infection with increasing antibody titers at the moment of sampling and no shedding of *Leptospira* DNA had occurred yet, or most likely that they were chronic carriers with falling or steady antibody titers and non-continuous *Leptospira* shedding in the urine. Similar findings have been previously described [14,15,18,20].

According to the World Health Organization’s guidance [24], the diagnostic panel of antigens used in MAT should include local strains, which increases the sensitivity of the technique compared to reference strains. However, the range of serovars should not be limited to local strains as the infection may be caused by a rare serovar or a strain not previously described. For this reason, we included saprophytic strains (*L. biflexa* strain Patoc I, *L. biflexa* strain CH 11, and *L. meyeri* strain Veldrat Semarang 173) in our panel diagnostic, which can cross-react with the antibodies generated by some pathogenic serovars. Possible explanations for the fact that none of the cats with antibodies against pathogenic leptospires had antibodies against the saprophytic serovars included in our MAT panel, could be that they have not either specific cross-reactivity against the used saprophytic ones or because as it has been described previously, the saprophytic serovars, specifically serovar Patoc, has limited ability to detect cross-reactions with antibodies of past infections [47].

The presence of leptospires in cats’ blood has been reported in clinical cases and in epidemiological research [6,10]. In our study, isolation of pathogenic *Leptospira* DNA in blood was only possible in 1 out of 89 cats. This cat was negative to FeLV/FIV tests, was seronegative against *Leptospira* and did not shed pathogenic leptospires DNA by urine. Therefore, we conclude that the animal was at an early stage of leptospirosis, with clinical signs non-present.

One epidemiological study in Taiwan, using serum and urine samples of cats [6], reported prevalence by PCR of 19.1% in blood and 67.8% in urine respectively. The differences in prevalence between this study and ours may be due to several factors such as i) Different primers used between studies. Chan et al., 2014 [6] used two sets of primers, the first one *Leptospira* rrs (16S) which is not able to differentiate between non-pathogenic *Leptospira biflexa* and pathogenic *Leptospira spp.* The second one primer set G1/G2 amplified a 285-bp sequence by PCR from strains of all pathogenic *Leptospira spp*. except for *Leptospira kirschneri.* This fact leads to differences between sensitivity and specificity in the PCR techniques used between studies. ii) The origin of the cats sampled in Taiwan; most of them came from rural areas; iii) The climatic conditions related to Taiwan, where typhoons are frequent and therefore favor the conditions for the maintenance of the leptospires in the environment; and iv) They used a random sample of cats which included shelter and household cats.

The prevalence of urinary shedding from leptospiral DNA in the present research is consistent with that reported in previous studies in cats [6,7,14,15,18,19]. Our results also match with those obtained by Sprißler et al., 2018 (0.8%) [14] and Gomard et al., 2019 (0.6%) [8]. In common, the two PCRs methodologies used in these two studies and in our work are targeting the *lipL32* gene [25,48]. In a sample of healthy cats and cats suffering from kidney disease, Rodriguez et al., 2014 [20] described a urinary prevalence of 14.9% in the latter group. Our results are not comparable with those, as the animals in our study were part of a free-roaming or shelter neutering program and at the time of sampling, none of the 4 cats showed clinical signs of disease. In cats, the urinary shedding time of *Leptospira* DNA remains unknown. Urine cultures of the cats in our study were not carried out due to the cumbersome and time-consuming nature of the methodology. The fact that none of the urinary shedding cats had antibody titers by MAT, may indicate that they have been chronically infected animals with falling, steady, or non-present antibody titers at the moment of sampling [35,36]. The MAT test has limitations detecting renal carriers [23]. The panel used in our study was broad, 8 species of *Leptospira*, belonging to 20 serogroups, 27 serovars, and 28 strains even including non-pathogenic serovars, so we can rule out the possibility that the *Leptospira* infection was sub-diagnosed.

One limitation of the research was that it was not possible to perform a follow up of tests of the cats, in order to determine seroconversion and the urinary shedding of pathogenic *Leptospira* DNA over time. This was due to the cats’ origin, as mentioned above.

## 5. Conclusions

In conclusion, this study reports seropositivity by MAT, against serovars belonging serogroups Australis, Autumnalis, Bataviae, Cynopteri, Grippotyphosa, and Pomona, for the first time in cats from Spain. Moreover, the antibodies against serovar Cynopteri (serogroup Cynopteri) were the most frequently detected in cats from Cáceres (Extremadura, southwest Spain). Knowledge of the involved leptospiral serovars in animals from any country is imperative for the accurate diagnostics and epidemiological surveillance of the disease. Therefore, we recommend including at least the aforementioned serogroups in any MAT panels used by diagnostic laboratories for detecting leptospiral antibodies in cats as well as dogs from Spain.

To our knowledge, this is the first report of urinary shedding of pathogenic leptospiral DNA in cats from Spain, diagnosed by means of molecular tools targeting the *lipL32* gene. Free-roaming cats in Spain can shed *Leptospira* DNA in their urine and may be a source of infection for people. Although the presence of viable leptospires in urine culture from cats has already been shown (p. 227, [19]), more prospective studies should be performed to ascertain the role of cats in the spread of the zoonosis.

## Figures and Tables

**Table 1 ijerph-17-01600-t001:** Species, serogroup, serovar, and strain from *Leptospira* tested among 244 cats from Spain.

SPECIES (8)	SEROGROUP (20)	SEROVAR (27)	STRAIN (28)
L. biflexa ^*^	Andaman	Andaman	CH 11
*L. interrogans*	Australis	Australis	Ballico
*L. interrogans*	Australis	Bratislava	Jez Bratislava
*L. interrogans*	Autumnalis	Rachmati	Rachmat
*L. borgpetersenii*	Ballum	Ballum	Mus 127
*L. interrogans*	Bataviae	Bataviae	Swart
*L. interrogans*	Canicola	Canicola	Hond Utrecht IV
*L. weilii*	Celledoni	Celledoni	Celledoni
*L. kirschneri*	Cynopteri	Cynopteri	3522 C
*L. kirschneri*	Grippotyphosa	Grippotyphosa	Mandemakers
*L. kirschneri*	Grippotyphosa	Grippotyphosa type Moska	Moskva V
*L. interrogans*	Hebdomadis	Hebdomadis	Hebdomadis
*L. interrogans*	Icterohaemorrhagiae	Copenhageni	Wijnberg
*L. interrogans*	Icterohaemorrhagiae	Icterohaemorrhagiae	Kantorowic
*L. borgpetersenii*	Javanica	Poi	Poi
*L. borgpetersenii*	Mini	Mini	Sari
*L. noguchii*	Panama	Panama	CZ 214
*L. interrogans*	Pomona	Pomona	Pomona
*L. noguchii*	Pomona	Proechimys	1161 U
*L. interrogans*	Pyrogenes	Pyrogenes	Salinem
*L. borgpetersenii*	Sejroe	Hardjo type bovis	Sponselee
*L. interrogans*	Sejroe	Hardjo type prajitno	Hardjoprajitno
*L. borgpetersenii*	Sejroe	Saxkoebing	Mus 24
*L. borgpetersenii*	Sejroe	Sejroe	M 84
*L. biflexa^*^*	Semaranga	Patoc	Patoc I
*L. meyeri* ^*^	Semaranga	Semaranga	Veldrat Semarang 173
*L. santarosai*	Shermani	Shermani	1342 K
*L. borgpetersenii*	Tarassovi	Tarassovi	Perepelitsin

* Nonpathogenic leptospires. All others are pathogenic.

**Table 2 ijerph-17-01600-t002:** Description of the sequence of *lipL32,* internal set primers, probe, and synthetic internal control used in the study.

Oligo ID	Sequence	Sequence Source
LipgrF2	5’CGCTGAAATGGGAGTTCGTATGATTTCC3’	*lipL32*
LipgrR2	5’GGCATTGATTTTTCTTCYGGGGTWGCC3’	*lipL32*
LipgrP1	5’FAM AGGCGAAATCGGKGARCCAGGCGAYGG3’BHQ1	*lipL32*
IntoF2	5’TAGAATCATTGAATCTATCACATCTCATG3’	Internal Control
IntoR2	5’TTGAACTAAATGTAGACTAAAGATGATCG’3	Internal Control
IntoP1	5’TxRd TTCACATTAACATTCAATAATCAATCATGAA3’BHQ2	Internal Control
PlasintS1	5’CTATAGAATCATTGAATCTATCACATCTCATGTACTTCACATTAACATTCAATAATCAATCATGAATTAATTCAATTTCTGATATGAATCGATCATCTTTAGTCTACATTTAGTTCAATATATC3’	Internal Control artificial template

**Table 3 ijerph-17-01600-t003:** Seropositive Microscopic agglutination test (MAT) results among 244 cats tested in Spain.

Gender	Age y.o.	Origin	Titer	Species	Serogroup	Serovar	Strain	FIV	FeLV
F	1	C	1:20	*L. borgpetersenii*	Ballum	Ballum	Mus 127	N	N
M	0.5	C	1:20 1:20	*L. interrogans* *L. kirschneri*	Australis Cynopteri	Bratislava Cynopteri	Jez Bratislava 3522 C	N	N
M	1	C	1:20	*L. kirschneri*	Cynopteri	Cynopteri	3522 C	N	N
M	2	C	1:20	*L. kirschneri*	Cynopteri	Cynopteri	3522 C	N	N
M	2	C	1:40	*L. kirschneri*	Cynopteri	Cynopteri	3522 C	N	N
M	2	C	1:320	*L. interrogans*	Bataviae	Bataviae	Swart	N	N
M	5	C	1:20	*L. borgpetersenii*	Ballum	Ballum	Mus 127	N	N
F	1	C	1:20 1:20 1:80 1:40 1:20 1:80 1:320	*L. interrogans* *L. kirschneri* *L. kirschneri* *L. kirschneri* *L. interrogans* *L. noguchii* *L. interrogans*	Australis Cynopteri Grippotyphosa Grippotyphosa Pomona Pomona Autumnalis	Bratislava Cynopteri Grippotyphosa Grippotyphosa-M Pomona Proechimys Rachmati	Jez Bratislava 3522 C Mandemakers Moskva V Pomona 1161 U Rachmat	N	N
M	2	C	1:20 1:20	*L. interrogans* *L. noguchii*	Pomona Pomoma	Pomona Proechimys	Pomona 1161 U	P	N
nM	7	B	1:20	*L. borgpetersenii*	Sejroe	Sejroe	M 84	N	N

F: Female; M: Male; nM: Neutered male; C: Cáceres, Extremadura; B: Barcelona, Catalonia; N: Negative; P: Positive, FeLV: feline leukaemia virus, FIV: feline immunodeficiency virus.

**Table 4 ijerph-17-01600-t004:** Results of DNA detection in blood and urinary shedding by polymerase chain reaction (PCR) in cats from Spain.

Gender	Age y.o.	Origin	Blood DNA Amplification by PCR (*n* = 89)	Urine DNA Amplification by PCR (*n* = 232)	FIV	FeLV
F	1	C	N	N	N	N
M	0.5	C	N	N	N	N
M	1	C	N	N	N	N
M	2	C	N	N	N	N
M	2	C	N	N	N	N
M	2	C	N	N	N	N
M	5	C	N	N	N	N
F	1	C	N	N	N	N
M	2	C	N	N	**P**	N
nM	7	B	N	N	N	N
M	0.5	C	N	**P**	N	N
F	1	C	N	**P**	N	N
M	0.5	B	N	**P**	N	N
F	0.5	B	N	**P**	N	N
F	0.6	B	**P**	N	N	N

F: Female, M: Male, nM: Neutered male, y.o: Years old, C: Cáceres, B: Barcelona, U PCR: Urinary PCR, B PCR: Blood PCR, N: Negative, P: Positive.

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
