# Peer review of "Leptospira Detection in Cats in Spain by Serology and Molecular Techniques"

_ijerph, 2020, doi:10.3390/ijerph17051600_

Round 1
Reviewer 1 Report
Thank you. I enjoyed reading the paper. Cats seem to have been an under-studied host species re leptospirosis. Generally your paper seems scientifically sound. There are however a number of queries raised for the reader. This may not be a problem with methodology so much as a failure to fully communicate your methods clearly and fully. This will become apparent from some of the questions raised below which I would appreciate your response to.
Line 49: It states that bacterial DNA isolate from acute human cases of lepto showed similarities to those isolated from cats. There is no reference for this. Was the similarity at the species level, serovar or WGS etc.? Without this information it is difficult to assess what weight to give that statement.
Line 54-55: That paper [22] stated that that observation may be biased as cat ownership was often related to trying to keep rat numbers down around the home so, not really independent factors. I would recommend removing that sentence.
Line 107-108: This reference is not published yet so the reader cannot refer to it. Was there any data for limits of detection, positive controls etc?Performance/efficiency of PCR on blood vs urine samples? Did you perform replicates or duplicates?
Table 1. L. weili should be italicised.
You state that residual samples of blood were used for the this study therefore ethics was not required. Presumably the blood was collected for FeLV/FIV testing and is part of the normal practice for the neutering programme. However, the authors state that urine was collected by cystocentesis using sterile needles whilst the cats were under anaesthesia. Was there ethics approval for this please?
Table 3 lacks clarity. The data for the 0.5 year old male and those of the 2 year old male at the bottom of the table do not align across the table making it difficult to see what data are attributable to that case. I would also include titre values for each serovar as it is not clear if the reader is meant to interpret the missing titre as being identical to the one above.
In Table 3 you have readings of 1:20 L. interrogans Pom, Pom and L. noguchii 1:80 for one of the female cats. Given both serovars belong to the same serogroup, do you think the lower titre, 1:20, could just represent a cross-reaction with the higher serovar titre reading? Also, and this applies throughout the text, 1:20 is a reciprocal titre, 20 is the titre.
Also, it was stated that the L. biflexa strains Patoc I and CH 11 were tested. These are used as a non-specific screening for MAT positivity for lepto yet there is no further mention of these serovars in the paper. Presumably all the MAT positives in Table 3 should have been positive for these too. Please explain their inclusion/use/results.
Line 178: It is not clear how "an increasingly individualized society" links with increased cat ownership. Maybe a different choice of word or an explanation would clarify this.
Line 255: If you all used a similar PCR method but your reference for this [25] has not been published yet, then you should use one of the other published papers to reference the method.
Author Response
Please see the attachment,Thank you.

Reviewer 2 Report
Overall, this is very well done and informative.
Lines 49-50 in the introduction mention that sequences from bacterial DNA from human cases have shown similarities to those from cats but I don't see a citation for this information.
In the Materials and Methods section under 2.1, the last sentence indicates that residual samples of blood were used and therefore ethical approval was not required. However, I don't see any ethical approval obtained for the cystocentesis procedures that were performed to obtain the urine samples. This procedure seems even more invasive and it seems as though there should be some ethical oversight. I recommend clarification.
In the Discussion section, lines 232-235 indicate a possible explanation for why cats who are seropositive may not have Lepto DNA present in their urine being that the infection is too acute (this is the opposite to what is stated in lines 262-264, which seems more correct). I would think that the opposite would make more sense--that cats with negative urine PCR but Lepto antibodies would instead have a slightly more chronic infection. This would be a good place to include information about what is currently known regarding the pathogenesis of Lepto in cats and time to mount an antibody response following infection versus time to develop a bacteremia versus time to begin shedding in their urine. If the details are not yet well-known for cats, similar information that is known for dogs or other species should be mentioned here to help the reader better understand the results from your study.
In lines 241-242, it is unclear to me what test was run on serum for this study in Taiwan--MAT or PCR? This is very important to the understanding of the paragraph and it's not clear.
Line 248-9 is confusing and I would encourage revision of the sentence ii) A third of the sampled cats by them, belonged to countryside.
Author Response
Please see the attachment, thank you.
